# Supramolecular Nature of Multicomponent Crystals Formed from 2,2′-Thiodiacetic Acid with 2,6-Diaminopurine or N9-(2-Hydroxyethyl)adenine

**DOI:** 10.3390/ijms242417381

**Published:** 2023-12-12

**Authors:** Jeannette Carolina Belmont-Sánchez, Duane Choquesillo-Lazarte, María Eugenia García-Rubiño, Antonio Matilla-Hernández, Juan Niclós-Gutiérrez, Alfonso Castiñeiras, Antonio Frontera

**Affiliations:** 1Department of Inorganic Chemistry, Faculty of Pharmacy, University of Granada, 18071 Granada, Spain; carol.bs.quimic@hotmail.com (J.C.B.-S.); jniclos@ugr.es (J.N.-G.); 2Laboratorio de Estudios Cristalográficos, IACT, CSIC-Universidad de Granada, Av. de las Palmeras 4, 18100 Armilla, Granada, Spain; duane.choquesillo@csic.es; 3Department of Physical Chemistry, Faculty of Pharmacy, University of Granada, 18071 Granada, Spain; rubino@ugr.es; 4Department of Inorganic Chemistry, Faculty of Pharmacy, University of Santiago de Compostela, 15782 Santiago de Compostela, Spain; 5Department de Química, Universitat de les Illes Balears, Crta. de Valldemossa km 7.5, 07122 Palma de Mallorca, Spain; toni.frontera@uib.es

**Keywords:** 2,2′-thiodiacetic acid, 2,6-diaminopurine, N9-(2-hydroxyethyl)adenine, crystal structures, molecular salts, proton transfer prediction

## Abstract

The synthesis and characterization of the multicomponent crystals formed by 2,2′-thiodiacetic acid (H_2_tda) and 2,6-diaminopurine (Hdap) or N9-(2-hydroxyethyl)adenine (9heade) are detailed in this report. These crystals exist in a salt rather than a co-crystal form, as confirmed by single crystal X-ray diffractometry, which reflects their ionic nature. This analysis confirmed proton transfer from the 2,2′-thiodiacetic acid to the basic groups of the coformers. The new multicomponent crystals have molecular formulas [(H9heade^+^)(Htda^−^)] **1** and [(H_2_dap^+^)_2_(tda^2−^)]·2H_2_O **2**. These were also characterized using FTIR, ^1^H and ^13^C NMR and mass spectroscopies, elemental analysis, and thermogravimetric/differential scanning calorimetry (TG/DSC) analyses. In the crystal packing the ions interact with each other via O–H⋯N, O–H⋯O, N–H⋯O, and N–H⋯N hydrogen bonds, generating cyclic hydrogen-bonded motifs with graph-set notation of $R_{2}^{2}$(16), $R_{2}^{2}$(10), $R_{3}^{2}$(10), $R_{3}^{3}$(10), $R_{2}^{2}$(9), $R_{3}^{2}$(8), and $R_{4}^{2}$(8), to form different supramolecular homo- and hetero-synthons. In addition, in the crystal packing of **2**, pairs of diaminopurinium ions display a strong anti-parallel π,π-stacking interaction, characterized by short inter-centroids and interplanar distances (3.39 and 3.24 Å, respectively) and a fairly tight angle (17.5°). These assemblies were further analyzed energetically using DFT calculations, MEP surface analysis, and QTAIM characterization.

## 1. Introduction

The final objective of so-called crystal synthesis is the preparation of crystals by design, for which it is necessary to understand the structure–property relationships, as well as the ability to control the molecular assembly in the programmed structure. For this, the contribution must be taken into account, both to the final structure and its properties, of inter- and intra-molecular interactions based specifically on hydrogen bonds and π–π stacking interactions, among others [1]. In this context, crystal engineering and covalent synthesis based on supramolecular synthons are essential tools for the design and synthesis of multicomponent crystals, which from a scientific point of view can be classified into solvates, salts, and co-crystals [2]. Along these lines, in the rational design of co-crystals or molecular salts, particularly in multicomponent crystals that contain acidic and basic constituents and where hydrogen-bond interactions are present, an empirical rule is usually applied by which it is established that when Δp*K*_a_ = [p*K*_a_(base) − p*K*_a_(acid)] > 4, the process typically results in salt formation, and if Δp*K*_a_ < −1 then co-crystals are more likely to form. When Δp*K*_a_ ranges between −1 and 4, either of the two systems is potentially viable. [3]. This concept is referred to as the salt-co-crystal continuum [4]. Numerous organic acids are very frequently used in the preparation of multicomponent crystals, especially as pharmaceutical co-crystal formers [5]. However, their use is not limited to this purpose. Conversely, many organosulfurs serve as an important tool in the study of many materials [6]. The fusion of these functional group types presents an opportunity to introduce novel coformers for synthesizing alternative solid forms, thus becoming an important method to alter the physicochemical properties of functional organic materials. The crystalline form of a compound significantly influences the properties of these materials and can impart changes to the crystal’s thermal and mechanical stability. In this context, it is important to note that the C-S bond is prevalent in numerous natural products, drugs, proteins, and advanced materials [7]. Compounds containing carboxylic groups and specific thioether functional group exhibiting C-S-C connectivity could be appealing in the deliberate design of crystal synthesis.

2,2′-Thiodiacetic acid (H_2_tda) stands as a versatile dicarboxylic organosulfur compound, and has been extensively investigated for its multidentate coordination and chelating capabilities with various transition metals. Some studies have revealed its efficacy as a potent antibacterial and cytotoxic agent. [8]. However, research in the synthesis of multicomponent crystals involving this acid remains limited. Molecular crystals in a 1:1 ratio have been documented with 1,7-phenanthroline [9], iso-nicotinamide [10], 4,4′-bipyridyl [11,12], and trans-1,2-bis(4-pyridyl) ethylene [13]. Yet, to date, only charge transfer complexes of thiodiacetic acid with ethylenediamine and o-phenylenediamine have been reported [14].

In the present article, 2,2′-thiodiacetic acid (H_2_tda) was selected as a multicomponent crystal former to be combined with two purine derivatives, N9-(2-hydroxyethyl)adenine (9heade) and 2,6-diaminopurine (Hdap, also known as 2-aminoadenine) (Figure 1), through non-covalent hydrogen bonds. These organic molecules have several hydrogen atoms available for donation and consist of five atoms each, and are thereby capable of serving as hydrogen-bond acceptors.

9heade is a synthetic purine nucleoside that has long demonstrated moderate antitumor activity when formulated as 2′-phosphonates [15]. Current research trends in this field strongly support the well-documented concept that such nucleotide analogs serve as excellent templates for the drug design due to the absence of the labile glycosidic bond. Additionally, the stability of the phosphonate moiety prevents easy enzymatic or chemical hydrolysis. Recent advancements in this realm have revealed that acyclic nucleoside phosphonates carrying the adenine nucleobase effectively inhibit Trypanosoma brucei adenine phosphoribosyltransferase in vitro [16].

Conversely, it is widely recognized that Hdap holds significant pharmaceutical interest, as enzymatic oxidative deaminations transform it in guanine [17]. This characteristic presents opportunities for its use as a prodrug in important antiviral medications such as acyclovir, penciclovir, or entecavir. Furthermore, recent breakthroughs have associated Hdap with pivotal stages in comprehending the origins of life under anaerobic conditions with abundant UV radiation. This association is attributed to the formation of functional and photostable RNA/DNA oligomers [18].

In this context, the objective of this study was to prepare, characterize, and acquire information regarding the protonation state of multicomponent crystals formed by H_2_tda with 9heade and Hdap. To comprehensively understand the properties of the resulting crystals, we conducted analyses using FTIR spectroscopy and single-crystal X-ray diffraction. Additionally, we investigated their stability through differential scanning calorimetry (DSC) and thermogravimetric analysis (TGA). Moreover, we quantified the role of both inter- and intra-molecular interactions through density functional theory (DFT) calculations, molecular electrostatic potential (MEP) surface analysis, and characterization using the “Atoms in molecules” theory (QTAIM).

## 2. Discussion and Results

The crystallization processes were conducted based on the p*K*_a_ values of each component. These values were predicted as 4.12 for H9heade^+^ [19], 4.98 for H_2_dap^+^ [20], and 3.13 for H_2_tda [21]. According to the pKa rule, the respective ΔpKa values (p*K*_a_(protonated base)–p*K*_a_(acid)) were 0.99 for (9heade)/(H_2_tda) and 1.85 for (Hdap)/(H_2_tda), clearly positioning them within the salt–co-crystal continuum. Commenting on experimental data obtained from salts/co-crystals extracted from the CSD [22], it has been observed that within the salt–co-crystal continuum region, there exists an empirical trend governing the probability of salt formation:*P*_obs_(salt, %) = 17 × Δp*K*_a_ + 28

Consequently, when applied to the systems currently under study, the likelihood of proton transfer is 44.8% for 9heade/H2tda and 59.5% for the Hdap/H_2_tda system. Working within this range might grant access to charged or neutral states. Given the distinct properties of salts and co-crystals, this outcome is likely influenced by other factors. Three crystal images of compound **1** are included in the Appendix A.

The dicarboxylic acid utilized in this study resulted in the formation of co-crystals or salts based on the Δp*K*a values. Salts were prepared by mixing solutions of both components in water and subsequently crystallizing the solution (see Experimental Section). H_2_tda formed salts with N9-(2-hydroxyethyl)adenine (9heade), and 2,6-diaminopurine (Hdap), yielding the salt [(H9Heade^+^)(Htda^–^), 1:1] **1** and the salt hydrate [(Hdap^+^)_2_(tda^2–^)·2H_2_O, 2:1:2] **2**. The structures of these crystalline multicomponent forms were determined using single crystal X-ray diffraction and FT-IR spectroscopy. The purity of the bulk phase was confirmed by thermal techniques (DSC and TGA). Additionally, DFT calculations were performed for these novel forms.

### 2.1. Structural Description and Supramolecular Analysis

The geometric parameters of H_2_tda, 9heade, and Hdap in compounds **1** and **2** align with those found in structures of their respective free compounds [23,24,25]. Hence, detailed discussion regarding these parameters will be omitted here. To analyze the crystal packing of **1** and **2**, the geometric parameters pertaining to the hydrogen-bond interactions are summarized in Table 1. Additionally, specifics regarding the π–π stacking interactions between aromatic rings in compound **2** are outlined in Table 2.

#### 2.1.1. [(H9Heade^+^)(Htda^–^), 1:1] Salt, 1

Colorless plate-shaped crystals of **1** were obtained through crystallization from water solution containing 9heade and H_2_tda in a 1:1 molar ratio. Analysis of their crystal structure reveals that the crystals belong to the triclinic space group *P*$\bar{1}$ (Table 1) with a H9heade^+^ cation and an Htda^–^ anion in the asymmetric unit (Figure 2a). Within the crystal structure, both ions form dimers (Figure 3a). In the cation, two adenine fragments are linked through two hydrogen bonds N6–H⋯N7^c^ and N6^c^–H⋯N7 (Table 2) forming a homosynthon of motif $R_{2}^{2}$(10) (Figure 3c). The distance of 2.928(2) Å lies between those found in the polymorph II structure, which range from 3.109 and 2.891 Å, formed by ribbons of 9Heade molecules [24]. On the other hand, the dimeric anion units form through robust hydrogen bonds O21–H⋯O11^f^ and nearly linear O21^f^–H⋯O11 (O–H⋯O, 179°) (Table 2), also giving rise to a homosynthon of graph-set $R_{2}^{2}$(16) (Figure 3d). Within this dimer the C–S–C bond angle measures 100.861(9)°, slightly less than the 95.8° found in the H_2_tda structure [23]. These dimeric units interconnect through additional hydrogen bonds. A carboxylic group and another carboxylate of two (Htda)^–^ entities interact, forming a linear homosynthon (O21–H⋯O11, 2.505 Å). This configuration acts as a double acceptor interacting with an aminopurine group functioning as a double donor, N6–H⋯O22^d^ and N1–H⋯O11^b^, leading to the formation of a graph-set heterosynthon $R_{3}^{2}$(10) (Figure 3e). Notably, the N1–H⋯O11^b^ bond demonstrates proton transfer from the carboxylate O11–C12–O12 to the N1 atom, which is the most basic nitrogen atom of the adenine residue in the 9head [26].

In the crystal packing, the cationic and anionic dimers arranged alternately forming ribbons along the “b” axis. These ribbons further interconnect along the “c” axis via an additional hydrogen bond, where the OH of the **1**-hydroxyethyl fragment participates as donor and the O12 of the carboxylate as acceptor (Figure 3f). This packing arrangement in the packing is not flat but resembles a ladder shape, with each rung comprising cationic and anionic dimers, evident in the projection onto the “ab” plane of **1** (Figure 3b). Several C–H⋯O (3.319–3.75 Å, 1135.5–172.7°) and C–H⋯S (3.748 Å, 130.6°) interactions involving methylene groups of both components reinforce the stability of the crystal structure.

#### 2.1.2. [(Hdap^+^)_2_(tda^2−^)·2H_2_O, 2:1:2] Salt Hydrate, 2

The crystallization of aqueous solutions containing Hdap and H_2_tda in a 1:1 molar ratio yields colorless block-shaped crystals, effectively maintaining the aforementioned molar ratio, constituting compound **2**. The crystal structure has been resolved and refined within the monoclinic space group *C2*/*c*, comprising an Hdap^+^ cation, a half tda^2–^ anion, and a water molecule for crystallization in the asymmetric unit (Figure 2b). Consequently, the compound’s formula is (Hdap)_2_(tda)·2H_2_O, with the two cations symmetrically related. The presence of water of crystallization molecules has probably prevented the expected formation of acid-pyridinamine heterosynthons involving O–H⋯N and N–H⋯O hydrogen bonds. The general crystalline structure can be considered as segregated by two well-differentiated substructures, one of them cationic formed only by Hdap^+^ cations, and an anionic one constituted by tap^2–^ anions and the two crystallization water molecules (Figure 4a).

The cationic substructure is straightforward, consisting of bands of coplanar Hdap+ cations along the “a” axis. These cations are interconnected through head-to-tail interactions via hydrogen bonds between an NH bond of each amino group (N2, N6) acting as donors and non-protonated nitrogen atoms (N3, N7) from adjacent cations serving as acceptors, thereby forming $R_{2}^{2}$(9) motifs (Figure 4b). This arrangement differs from the structures observed in both Hdap·H_2_O and in (Hdap)_2_^+^(hpt)^2–^·7H_2_O (hpt^2–^ = 2-(2-carboxylatophenyl) acetate) [19]. 

The anionic substructure, unlike in 1, is constructed in a manner where the bideprotonated anions tda^2–^ are interconnected by water molecules through hydrogen bonds. Here, the oxygen atoms O12 of both carboxylates act as acceptors, thereby creating graph-set $R_{4}^{2}($8) heterosynthons that alternate with the anions along the “a” axis (Figure 4b), forming a ladder-shaped arrangement. 

The cationic and anionic substructures are interconnected through additional hydrogen. In this arrangement, the protonated nitrogen atoms of each cation, along with one of the two amino N-H groups that are not involved in the cationic bond, serve as donors towards hydrogen atoms from water molecules (N1–H⋯O1^b^). Furthermore, they interact with other carboxylate oxygen atoms from different equivalent positions (N6–H⋯O11^d^ and N9–H⋯O11^a^), as detailed in Table 1. These interactions result in the formation of two distinct heterosynthons characterized by ring motifs $R_{3}^{3}$(10) and $R_{3}^{2}$(8), as illustrated in Figure 4b. Consequently, the anionic substructure connects two planar cationic substructures in a manner resembling an open book, forming an angle of 99.9(4)°, which corresponds to the C-S-C bond ring. This configuration depicts the thioether atom functioning as a pivotal hinge, as depicted in Figure 4a.

The final result is a three-dimensional network where, along axis “b”, the substructures are interleaved (Figure 4a). However, observed in the “ac” plane, it resembles a laminated network alternately linked by S-atoms (Figure 5a) and by aromatic ring–ring π,π-stacking interactions, between bilayers, in the cationic substructure (Figure 5b) with inter-centroid distances of 3.48 or 3.49 Å (Table 2) but an interplanar dihedral angle of 0° between the involved purines at a distance of 3.23 Å.

### 2.2. IR Spectra

In this section all wavenumbers are given in cm^−1^. It has been shown that deprotonation of a carboxylic acid to form a carboxylate not only equalizes the C=O bond length, but also has a significant effect on the C–O/C=O strain vibrations observed in IR spectra. Thus, while the carboxylates give rise to two carbonyl strain bands, one strong asymmetric near 1600 and the other weakly symmetric near 1400 [27], the carboxylic acids in the co-crystals show strong C–O bands above 1600 (near 1700), along with weak C=O bands near 1275 confirming a non-ionized state of the adduct [28].

An analysis of the IR spectra obtained for both compounds (Appendix A) shows the presence of broad bands at 1950 and 2450, indicating the presence of O–H⋯N hydrogen-bond interactions [29]. This broadness hinders the observation of the typical series of weak peaks related to the stretching mode(s) of N^+^–H (2800–2250).

We can anticipate that the IR spectrum of **1** (Appendix A) will be rather difficult for tentative assignations of bands to specific stretching and bending modes. A convenient manner to interpret the IR spectra of the compound reported here is looking at the IR spectra of the corresponding N components The spectrum of **1** revealed the presence of H9heade^+^ with the bands of ν_as_ (3347), ν_s_ (3286), and δ (1672) of –NH_2_, ν (3183) and δ (1510) of N9–H, broadness peaks at 2800–2250 (among other peaks at 2687, 2361), and δ (1548) of N1^+^–H and ν (1071 or 1044) of ν (C–O) (both for the C−O(H) in H9heade^+^ or in Htda^−^). Note that the ν (O–H) of both H9heade^+^ and Htda^−^ are not identified. That, as well as the broadness of the peaks related to the stretch of N^+^–H (also rather typical in IR spectra of tertiary amine hydrochlorides), are clearly related to the hydrogen bonding, and lower frequencies (and so wavenumbers) and broad bands. Out-of plane deformations, π (C2–H) and π (C8–H), are related to defined peaks at 872 and 852 (usually in the range 900–860). In addition, the anion Htda^−^ produces the ν (C=O) band at 1702, ν_as_ (1618), ν_s_ (1382) of –COO^−^, and the in-plane δ (O–H) 1236 of –COOH.

The IR spectrum of **2** (Appendix A) revealed intense broadening of absorptions in the range 3600–1300, due to the N–H⋯N, N–H⋯O, and O–H⋯O interactions. However, a few features are remarkable as follows. First, the ν_as_ (H_2_O) at 3448 is clearly identified; second, the broad ν_as_ (NH_2_) 3320 and the overlap contribution of ν_s_ (NH_2_) + ν_s_ (H_2_O) 3209; third, the series of peaks in the region 2800–2250 (typically at 2700~2250 [28,29]) related to the stretching of N^+^–H for tertiary amine hydrochlorides, here evidencing the N1^+^–H bond of the tautomer 2,6-(N1,N9)diaminopurinium(1+) cation, proved by the crystallographic results for **2**. The out-of-plane π (C–H) weak bands, expected at 900–860, are observed at 882 and 865; and fourth, two in-plane δ (N–H) absorptions are observed at 1558 (sh) and 1508 (w). Overlapped ν_as_ (COO) 1598 and 1577 absorptions, and with ν_s_ (COO) 1390, are the clearest evidence of the dianion tda^2−^.

### 2.3. ^1^H and ^13^C Spectroscopic Analysis

The ^1^H and ^13^C NMR spectra of [(H9Heade^+^)(Htda^–^), 1:1], **1**, and the salt hydrate [(Hdap^+^)_2_(tda^2–^)·2H_2_O, 2:1:2], **2,** were recorded in D_2_O or DMSO and are shown in Appendix A. In the ^1^H NMR of **1**, signals due to the OHs of Htda^−^ and H9heade^+^ are not observed because they are involved either in an acid–base proton transfer or in a strong hydrogen bond with the oxygen atoms of the carboxylic groups (see Table 1). In **2**, the protons of both tda^2−^ carboxylic groups have been transferred to two Hdap^+^ molecules, which engage in strong hydrogen bonds with the hydration water molecules, and the corresponding signals are also not observed. In the ^13^C NMR of **1**, the peak at 175.42 ppm is attributed to the carbonyl C atom of tda^−^ and the peaks at 46.47 and 35.10 ppm are due to the alkyl group of tda^−^. In **2**, the peaks at 177.86 and 177.83 ppm are attributed to the carbonyl C atoms of tda^2−^ and the peaks at 37.32 and 37.289 ppm are due at the alkyl groups in tda^2−^.

### 2.4. Salt–Co-Crystal Continuum

In the discussion surrounding the continuum between salt and co-crystal formations, apart from the methodologies reliant on the count and placement of IR bands associated with the carboxylic/carboxylate group and the estimation of the aforementioned Δp*K*_a_ value, there exist alternative prediction methods. These methods take into account specific structural parameters governing the synthesis of multicomponent crystals between carboxylic acids and bases, primarily influenced by hydrogen bonds.

(1)When carboxylic acids are used as conformers, an analysis of the C–O bond distances can distinguish between solid features of salt or co-crystal character. If both C–O distances differ by less than 0.03 Å the compound must be considered as a salt, whereas when one suspects a co-crystal structure, the distances C=O and C–OH within the carboxyl group differ by more than 0.08 Å [30]. In the compounds studied here, the distances for C=O and C–OH exhibit values of 1.235/1.221 and 1.283/1.308 for 1, and 1.224 Å and 1.231 Å for 2, respectively, resulting in differences of 0.048 and 0.087 Å for the two carboxylic groups of 1, and of 0.007 for 2. These values align with those found in salts containing carboxylic acids as coformers, indicating proton transfer from the acid to the base. However, at 1 the value found for the second carboxylic group, which is slightly higher than 0.08 A, is consistent with the absence of proton transfer and, consequently, the veracity of the method proposed by Gobetto et al. [30].(2)As an alternative approach to assessing the equilibrium between co-crystallization and proton transfer in a range of acid–base reactions, Aakeröy et al. [31] conducted a comparative analysis of crystal data involving a series of salts and co-crystals. Their study revealed that the average ratio of the carbonyl, C=O bond distance to the C–OH bond distance in co-crystals is 1.08, while the ratio of the C–O/C–O bond distance for the carboxylate anion is 1.02. Applying this concept to the compounds studied, the C–O/C–O ratios obtained are 1.039–1.071 Å for **1** and 1.005 Å for **2**. These values strongly support the inference that **2** is a salt, while **1** comprises deprotonated and neutral carboxylic groups, validating the assignment previously deduced from IR spectra and Δp*K*_a_ values for these systems studied. Furthermore, these values align consistently with calculations for other multicomponent crystals involving thiodiacetic acid as a coformer, as previously documented [8,9,10,11,12,13,14].

### 2.5. Thermal Analysis

TGAs offer insights into melting, crystallization, sublimation, decomposition, and solid-state transitions, and facilitate the observation and quantification of volatile compounds, including residual solvents and gaseous by-products. In this study we present the TGA behavior for both the novel co-crystal salts (Appendix A) and the utilized reagents (Appendix A).

The weight loss vs. temperature (°C) plots for **1** and **2** are shown in Figure 6. Compound 1 exhibits considerable stability against temperature variations. Its decomposition initiates within the range of 180–250 °C resulting in a weight loss of 7.34%. This stage involves the evolution of H_2_O and CO_2_, followed by a sharp step occurring between 250 and 465 °C, accounting for 51.12% of the weight loss. This step generates H_2_O, CO_2_, CO, SO_2_, H_2_C=CH_2_, and potentially unidentified gas, likely HSCN, alongside some N_2_O. Subsequently, a third step spanning 465–650 °C leads to a 40.19% weight loss, yields H_2_O, CO_2_, CO, and the typical three N-oxides (N_2_O, NO, and NO_2_) but notably no NH_3_. Interestingly, the TGA analysis of free 9heade does not indicate the presence of this gas (Appendix A). After these stages, a stable residue is formed, representing 1.16% or 1.03% at 650 or 950 °C, respectively. These data align with a partial overlapping combustion of both ligands, along with the anticipated stability of the purine moiety of 9heade.

Samples of compound **2** retain variable moisture and water content. They exhibit a sharp water loss at 135–180 °C, followed by the onset of organic thermal decomposition at 235 °C, occurring in two stages. The first stage spans 235–400 °C, resulting in a 27.85% weight loss, generating H_2_O, CO_2_, CO, likely HSCN, and SO_2_. Subsequently, between 400 and 800 °C, further decomposition occurs, yielding H_2_O, CO_2_, CO, NH_3_, N_2_O, NO, and NO_2_. At 800–900 °C, a stable residue of approximately 2% remains.

### 2.6. DSC Analysis

Appendix A depict the heat flow versus the temperature for compounds **1** and **2**, respectively. It is notable that the melting points of these compounds differ from both 2.2′-thiodiacetic acid (128–131 °C) and those of 9heade (242 °C) and Hdap (>300 °C). We observed that, for both 1 and 2, their melting point falls within the range of melting points of their respective coformers.

The DSC curve for **1** exhibits an endothermic peak onset at 188.50 °C, followed by two weaker peaks at 203.9 °C and 276.1 °C. The first peak corresponds to the salt fusion, while the subsequent peaks suggest a decomposition of compound **1** after/during the melting.

For **2**, the DSC curve is quite complex and we will only try to interpret the most significant peaks. The curve displays two distinct endothermic regions, possibly indicating partial dissociation among its components. The initial peak onset at 131.27 °C aligns well with the fusion point of 2.2′-thiodiacetic acid, albeit slightly shifted due to the heating rate of 10 °C/min. The subsequent maximum corresponds to Hdap, which decomposes/sublimates at 254.3 °C. The compound **2** showed broad and shallow twin peaks at 73.7 °C and 100.5 °C. The peak of dehydration is observed at around 73 °C (which is consistent with the TGA–IR results), while the small endothermic peak at 100 °C originates from another transition (maybe a polymorphic transition).

### 2.7. DFT Calculations

The theoretical study focuses on the energetic analysis of the H-bonded synthons described in Figure 3 and Figure 4. Additionally, we evaluated the π-stacking interaction between the protonated aza-adeninium rings that are important in the crystal packing of compound **2**.

To begin, we computed the MEP surfaces of the salts to investigate the relative H-bond donor/acceptor abilities of the coformers. In the case of compound **2**, we employed a model where the 2,2′-thiodiacetic acid was replaced by an acetate for simplicity. The MEP surfaces are depicted in Figure 7, revealing that in compound **1** the MEP minimum is located at the O-atom of the carboxylate group (−59.6 kcal/mol), followed by that at the carboxylic O-atom (−49.0 kcal/mol). The MEP values at the N-atoms of the adenine ring are notably smaller (−11.9 and −9.4 kcal/mol) due to the cationic nature of the adeninium ring. Conversely, the values at the NH_2_ group are notably high (53.9 and 45.2 kcal/mol) due to the protonation at N1. The MEP maximum is situated at the OH group of the pendant arm. In compound **2**, the MEP values at the carboxylate group are very large and negative (−94.1 and −81.5 kcal/mol), representing the MEP minima. Additionally, the MEP values are negative at the N-atoms of the aza-adeninium ring (−41.4 and −17.6 kcal/mol). The MEP maxima are located at the H-atoms of the water molecule that is H bonded to the N+–H group of the aza-adeninium ring (+94.1 and +87.8 kcal/mol). Lastly, the MEP values are notably positive at the NH_2_ groups, ranging from 67.9 to 81.6 kcal/mol. This MEP analysis anticipates the formation of strong H bonds and aligns well with the various of H-bonded synthons described above.

To assess the energies of the H bonds, we characterized the assemblies illustrated in Figure 8 using a combination of QTAIM and NCIplot methods. The strength of each H bond was computed using the reliable method proposed by Emaniam et al. [32], which calculates the association energy of the hydrogen bonds (HBs) based on the electron density (ρ) at the bond CP utilizing the equation E = −233.1 × ρ + 0.7. This method proves particularly useful for the systems under study in this manuscript (salts) where electrostatic effects predominantly govern (pure Coulombic attraction). Consequently, employing the QTAIM approach allows for estimation of the contribution of the H bond, independent of the ion-pair electrostatic force influence. The strength of each H bond is highlighted in Figure 9 (in red), unveiling that the O–H⋯O interactions that generate the $R_{2}^{2}$(16) synthon exhibit the highest strength. This aligns with the MEP analysis shown in Figure 7. Each H bond is characterized by a bond critical point (CP, depicted as a small red sphere) and a bond path (dashed bond) connecting the H to the O/N-atoms. Additionally, RDG isosurfaces correspond precisely with the bond CPs, displaying colors ranging from green for the weaker interactions to bluish for moderate ones and finally dark blue for the strongest ones. Regarding the $R_{2}^{2}$(16) synthon, an extended green isosurface appears between the COOH groups, suggesting the presence of weak van der Waals interactions.

The QTAIM/NCIplot analysis also evidences the existence of some weak CH⋯O,S interactions that were not described in Figure 3 and Figure 4 (marked with arrows in Figure 8b). The energies of the different synthons are also provided in Figure 8, evidencing that the $R_{2}^{2}$(16) and $R_{3}^{2}$(10) synthons are the most significant energetically because they involve the strongest H-bond donor and acceptor groups. The $R_{2}^{2}$(10) synthon connecting the adeninium rings by NH⋯N bonds is the weakest in line with the small MEP values at the N-atoms. In general, the large interaction energies of the synthons confirm the importance of these H bonds in the solid state of compound **1**, as described in Figure 3. 

A similar analysis was performed for compound **2**, as depicted in Figure 9. The H bonds are characterized by the corresponding bond CPs, bond paths, and RDG isosurfaces. The $R_{4}^{2}$(8) synthon involving the two water molecules is the most favorable (−28.5 kcal/mol). As also observed in **1**, the synthons involving OH⋯O and NH⋯O H bonds [$R_{4}^{2}$(8), $R_{3}^{3}$(8), and $R_{3}^{2}$(8)] are stronger than the $R_{2}^{2}$(9) synthon with two NH⋯N H-bonds between the aza-adeninium rings. The large binding energy of the $R_{4}^{2}$(8) synthon evidences the strong influence that the co-crystallized water molecules exert in the crystal packing of **2**, preventing the formation of additional heterosynthons.

Lastly, we assessed the π–π-stacking interaction in compound **2**, utilizing the model outlined in Figure 10. An evident strong complementarity of the aza-adeninium rings is observed, as the RDG isosurface encompasses the entire π-cloud of the rings, incorporating one of the exocyclic NH_2_ groups. The dimerization energy, calculated via the supramolecular approach, appears notably large and negative (−37.6 kcal/mol), likely influenced by ion-pair effects. This significant interaction energy, even surpassing that of the H-bonded synthons, reaffirms the pivotal role of the π–π stacking interaction in the crystal packing of compound **2**, as highlighted in Figure 5.

## 3. Materials and Methods

All reagents were purchased from Aldrich or TCI and used without any further purification. Analytical grade solvents were used for the crystallization experiments.

### 3.1. Instrumentation

Elemental analyses for carbon, hydrogen, nitrogen, and sulfur were performed with a Fisons-Carlo Erba 1108 microanalyser (Elemental Microanalysis Ltd., Okehampton, UK). Mass spectra were obtained on a Micromass AUTOAPEC spectrometer for ESI (Waters Cromatografía, S.A., Cerdanyola del Vallès, Spain). ^1^H NMR and ^13^C NMR spectra in DMSO-d_6_ or D_2_O were run on a Varian Mercury 300 instrument (Varian Medical Systems, Inc., Palo Alto, CA, USA), using TMS as the internal reference. (For the numbering of atoms, see Figure 2). The FT-IR spectra were recorded as KBr pellets (4000–400 cm^–1^) on a Jasco FT-IR 6300 spectrophotometer (Jasco Analítica, Madrid, Spain). TGA experiments were carried out on a Shimadzu Thermobalance TGADTG-50H Instrument (Shimadzu Europe GbmH, Duisburg, Germany) from room temperature to 950 °C in a flow of air (100 mL min^–1^). Additionally, a series of approximately 35 time-spaced FT-IR spectra per sample (including those of the own reagents) of evolved gases were recorded using a coupled FT-IR Nicolet Magna 550 spectrophotometer (Thermo Fisher Scientific Inc., Waltham, MA, USA). Differential scanning calorimetry (DSC) measurements were recorded for the samples (6.460 mg, **1**; 7.036 mg, **2**) on a DSC-SHIMADZU mod. DSC-50Q instrument (Shimadzu Europe GbmH, Duisburg, Germany) under an N2 atmosphere, at 26–400 °C (heating rate 10 °C/min).

### 3.2. Single-Crystal X-ray Diffraction

Diffraction data were obtained using a Bruker D8 VENTURE PHOTON III-14 diffractometer from crystals mounted on glass fibers. Corrections for Lorentz and polarization effects, as well as absorption, were applied using a multi-scan method [33].

The structures were solved by direct methods [34], which revealed the positions of all non-hydrogen atoms. These were refined on *F*^2^ by a full-matrix least-squares procedure using anisotropic displacement parameters [34]. Hydrogen atoms were located in the difference maps and the positions of O–H and N–H hydrogen atoms were refined (others were included as riders). Isotropic displacement parameters of H atoms were constrained to 1.2/1.5 *U*_eq_ of the carrier atoms. Molecular graphics were generated with DIAMOND software (Version 4.6.8) [35]. Crystal data, experimental procedures, and refinement outcomes are summarized in Table 3.

### 3.3. Preparation of Crystals **1** and **2**

Water solutions (50 mL) of 9Heade (15.7 mg, 0.1 mmol) with H_2_tda (16.6 mg, 0.1 mmol) or Hdap (22.2 mg, 0.1 mmol) with H_2_tda (16.6 mg, 0.1 mmol), were kept at room temperature. Upon slow evaporation of the solvent over about 5 days, colorless crystals **1**–**2** were obtained. 

#### 3.3.1. Crystals of [(H9Heade^+^)(Htda^–^)], **1**

Yield: 80% based on 9heade. IR (KBr, Pellet cm^−1^): 3347 (s), 3386 (s), 3076 (s), 2990 (w), 2960 + 2933 (m), 2874 (m), 2687 + 2361 (vw), 1961 + 1815 (w), 1702 (s), 1672 (s), 1618 (m), 1548 (w), 1410 (s), 1281 (s), 1382 (m), 1236 (s), 1071 (s), 872 (s). ^1^H NMR (500 MHz, DMSO, δ ppm) 8.26 (s, 1H, N1), 8.13 (s, 1H, C8), 8.06 (s, 1H, C2), 5.0 (s, 1H, O10), 4.2 (t, 2H, C9), 3.7 (t, 2H, C11), 3.3 (s, 4H, C10, C21). ^13^C NMR (126 MHz, D_2_O, δ ppm) 175.42 (C12, C22), 151.25 (C6), 148.73 (C2), 146.29 (C4), 144.55 (C8), 118.13 (C5), 59.66 (C10), 46.47 (C21), 35.10 (C9, C11). MS *m*/*z*, (%): 329 (M^+^, 24). Anal. calcd. for C_11_H_15_N_5_O_5_S (329.34): C, 40.12; H, 4.59; N, 21.26; S, 9.74; found C, 40.92; H, 5.05; N, 21.73; S, 9.52%.

#### 3.3.2. Crystals of [(Hdap^+^)_2_(tda^2–^)]·2H_2_O, **2**

Yield: 85%. IR (KBr Pellet, cm^−1^): 3448 (s), 3320 (sh), 3209 (br), 3071 (s), 2973 (br), 2896 (br), a series of 7 or more weak peaks at 2800–2250 (w), 1697 (s), 1653 (m), 1627 (m), 1598 (s), 1577 (sh), 1558 (sh), 1508 (sh), 1390 (s), 940 (s), 882 and/or 865 (w) ^1^H NMR (400 MHz, D_2_O, δ ppm) 7.70 (s, 2H; C8), 3.23 (s, 4H; C11). ^13^C NMR (101 MHz, D_2_O, δ ppm) 177.86 (C12), 177.83 (C12), 161.28 (C2), 158.07 (C6), 154.71 (C4), 151.47 (C8), 114.51(C5), 37.32 (C11), 37.28 (C11). MS *m*/*z*, (%): 485 (M^+^, 18). Anal. calcd. for C_14_H_22_N_12_O_6_S (486.49): C, 34.56; H, 4.56; N, 34.55; S, 6.59; found C, 35.35; H, 4.73; N, 34.10; S, 6.44%.

### 3.4. DFT Calculations

The energetic analysis of the hydrogen bonding and π–π stacking interactions were performed using the Gaussian-16 [36] and the PBE0-D3/def2-TZVP level of theory [37,38]. Crystallographic coordinates were employed to assess interactions in the solid state. The π–π stacking interaction in **2** was computed by determining the energy difference between isolated monomers and their assembly. Binding energies were assessed with correction for the basis set superposition error (BSSE) via the Boys–Bernardi method [39]. The Bader’s “Atoms in molecules” theory (QTAIM) [40] implemented using the AIMAll calculation package [41] was utilized to analyze and estimate the association energies of the H-bonding interactions discussed below. Molecular electrostatic potential surfaces (isosurface 0.001 a.u.) were computed using the Gaussian-16 software (Revision B.01) [36].

To analyze the nature of interactions in terms of attraction or repulsion and visualize them in real space, we utilized the NCIPLOT visualization index, which depicts the regions of reduced density gradient (RDG) [42] derived from the electronic density (ρ) [43]. The sign of the second Hessian eigenvalue (λ_2_) multiplied by the electron density (i.e., sign(λ_2_) ρ in atomic units) is employed to identify attractive/stabilizing interactions (blue–green-colored isosurfaces) or repulsive interactions (yellow–red colored isosurfaces) using 3D-Plots. The NCIplot index parameters used in this manuscript are as follows: RGD = 0.5; ρ cut off = 0.04 a.u.; color range: –0.04 a.u. ≤ sign(λ_2_) ρ ≤ 0.04 a.u.

## 4. Conclusions

The uncertainly within the Δp*K*_a_ values, typically ranging between −1 and 4, concerning the salt-co-crystal continuum and Δp*K*_a_ rules, was resolved in this instance. We achieved this by successfully obtaining stable multicomponent ionic crystals involving N^1^,N^9^-(2-hydroxyethyl)adeninium(1+) or N^1^,N^9^-2,6-diaminopurinium(1+) co-crystallized with 2,2′-thiodiacetic acid anions (Htda^−^ or tda^2−^, respectively). These crystals exhibit exhaustive N–H⋯N/N–H⋯O/ and O–H⋯O hydrogen bonds, giving rise to diverse supramolecular synthons. Analysis via single crystal X-ray diffraction unveiled that primary synthons between carboxylic acid and aminopurine moiety are absent. Instead, [(H9Heade^+^)(Htda^−^)] 1 exhibits a single-point O–H⋯O and a three-point supramolecular synthon of graph-set $R_{3}^{2}$(10), while [(Hdap^+^)_2_(tda^2−^)]·2H_2_O 2 features a single-point N–H⋯O and a three-point supramolecular synthon of graph-set $R_{3}^{2}($8), playing pivotal roles in the crystallization of these multicomponent crystals. DFT calculations and QTAIM analysis corroborated these observations, allowing for the evaluation of each H bond’s strength.

In general, OH⋯O and NH⋯O H-bonds involving both coformers and water molecules exhibit greater strength compared to the N–H⋯N between the adeninium(1+) and 6-aza-adeninium(1+) units. Notably, the FT-IR spectra and TGA of both studied compounds corroborate the crystallographic findings. This is evident in the stretching expression of the N^+^–H moiety and the heightened thermal stability observed in the water-free crystal **1**.

## Figures and Tables

**Figure 1 ijms-24-17381-f001:**
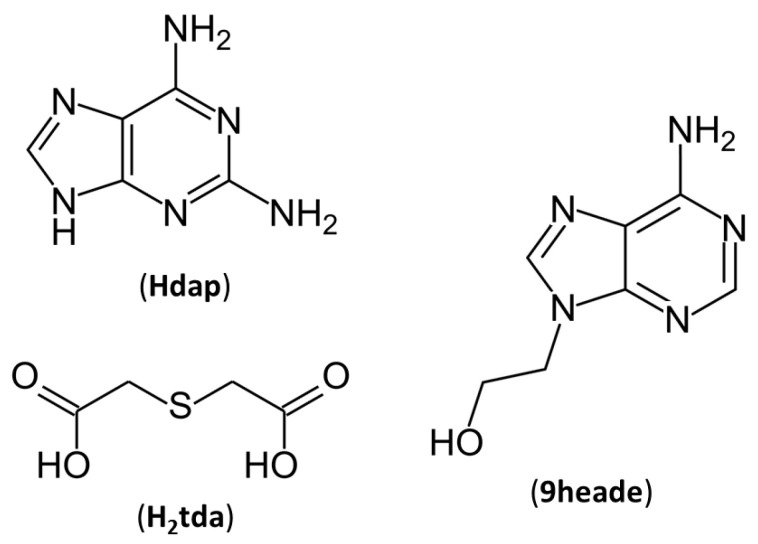
Chemical diagrams of, 2,6-diaminopurine (Hdap), 2,2′-thiodiacetic acid (H_2_tda) and N9-(2-hydroxyethyl)adenine (9heade).

**Figure 2 ijms-24-17381-f002:**
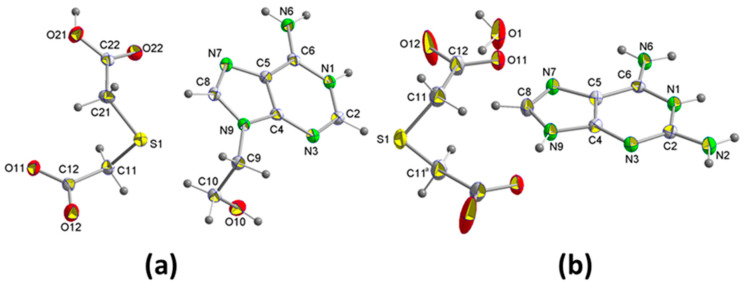
Perspective view of the asymmetric unit of: (**a**) [(H9Heade^+^)(Htda^−^)] **1** and (**b**) [(Hdap^+^)_2_(tda^2−^)]·2H_2_O **2** with atom labeling.

**Figure 3 ijms-24-17381-f003:**
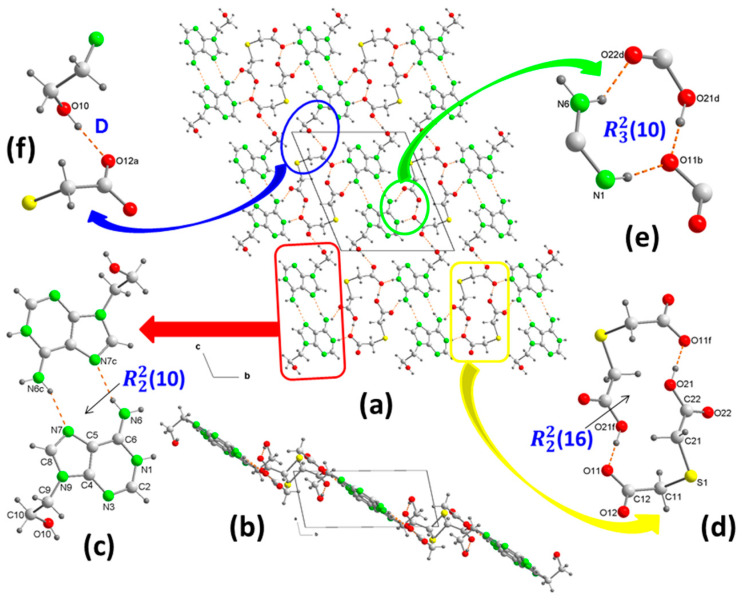
Crystal packing of the molecules in 1, projected onto the “bc” and “ab” planes (**a**,**b**, respectively). View of centrosymmetric ring motifs with a $R_{2}^{2}$(10) graph-set for (H9heade)^+^ (**c**), and a $R_{2}^{2}$(16) graph-set for (tda)^2−^ (**d**). View of the cation–anion intermolecular interactions and the supramolecular synthons (**e**,**f**). For symmetry codes see tables.

**Figure 4 ijms-24-17381-f004:**
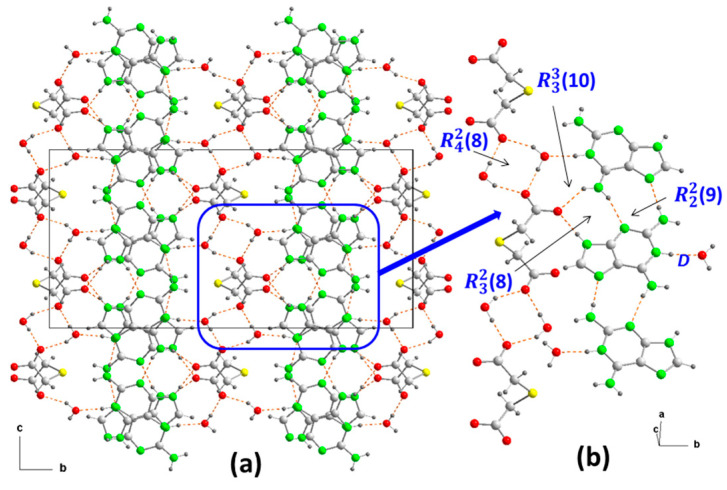
Crystal packing in **2** projected onto the “bc” plane, showing the intermolecular interactions (**a**) and the supramolecular synthons (**b**). For symmetry codes see tables.

**Figure 5 ijms-24-17381-f005:**
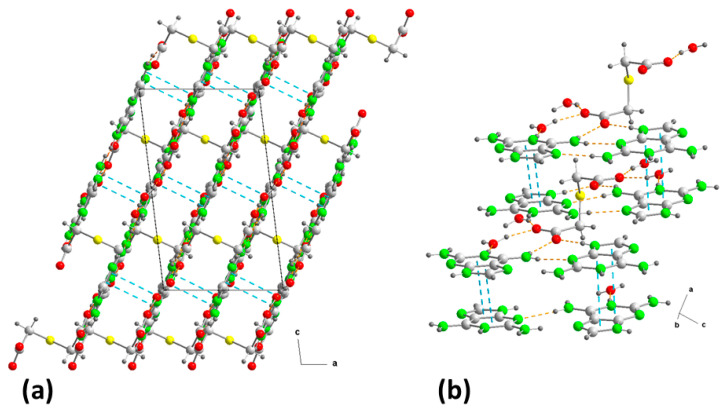
(**a**) Crystal packing in **2** projected onto the “ac” plane showing the layered structure along the b-axis. (**b**) Detail of the π–π stacking interactions in **2** along the “ca” direction. Hydrogen bonds are shown as orange dashed lines and ring–ring interactions as blue dashed lines. For symmetry codes see tables.

**Figure 6 ijms-24-17381-f006:**
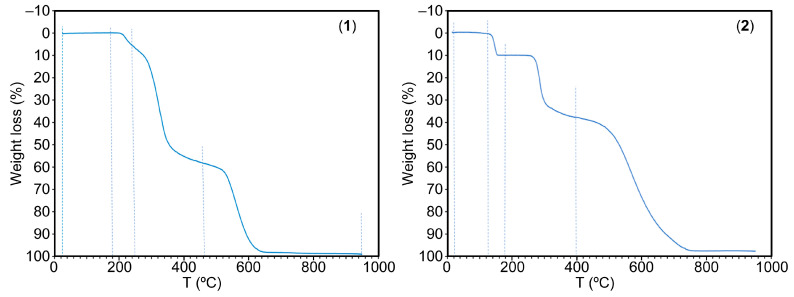
TGA curves of compounds **1** (**left**) and **2** (**right**).

**Figure 7 ijms-24-17381-f007:**
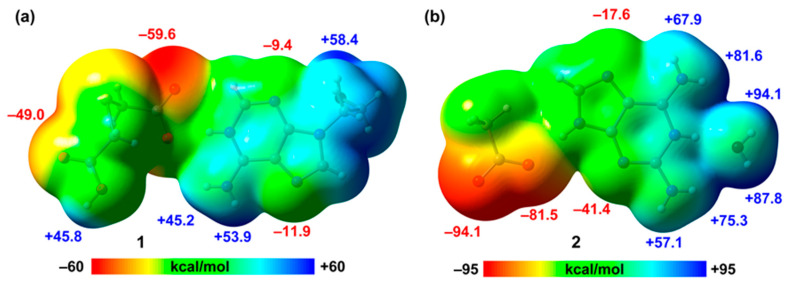
MEP surfaces of the salts of 1 (**a**) and a model of 2 (**b**) at the PBE0-D3/def2-TZVP level of theory (density isovalue 0.001 a.u.). The energies are given in kcal/mol.

**Figure 8 ijms-24-17381-f008:**
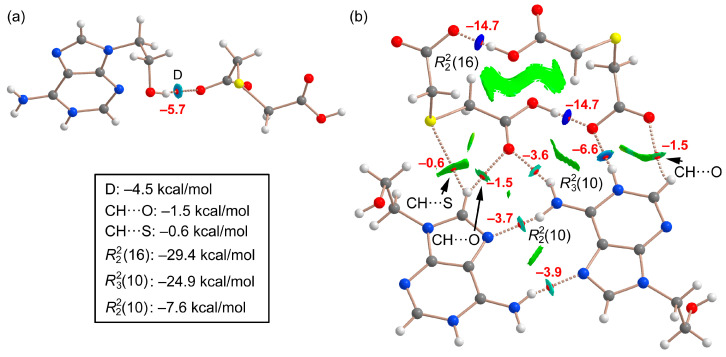
QTAIM/NCIPlot analysis of intermolecular bond CPs (red spheres), bond paths, and RDG isosurfaces of the H-bonded dimer (**a**) and tetramer of (**b**) compound **1**. The individual association energies of the H bonds are indicated using a red font next to the bond CPs.

**Figure 9 ijms-24-17381-f009:**
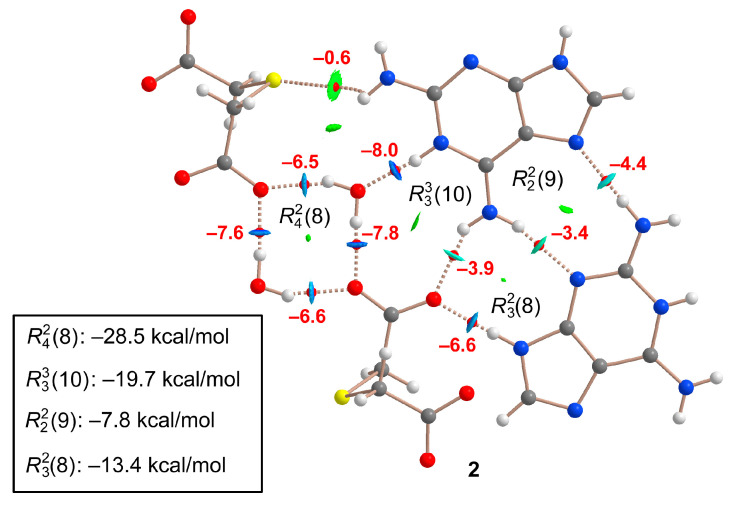
QTAIM/NCIPlot analysis of intermolecular bond CPs (red spheres), bond paths, and RDG isosurfaces of the H-bonded assembly of compound **2**. The individual association energies of the H bonds are indicated using a red font next to the bond CPs.

**Figure 10 ijms-24-17381-f010:**
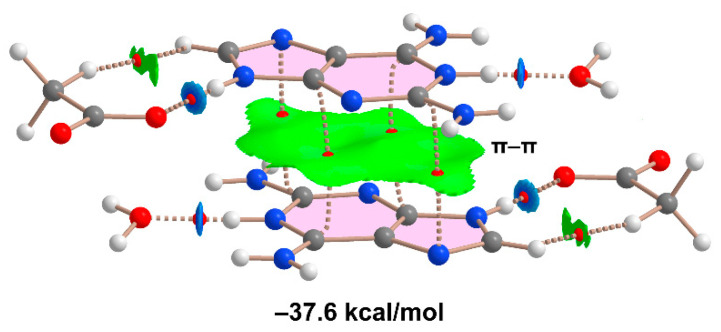
QTAIM/NCIPlot analysis of a model of the π–π-stacking assembly of compound **2** and the computed dimerization energy.

**Table 1 ijms-24-17381-t001:** Hydrogen-bond parameters [Å, °] for [(H9Heade^+^)(Htda^−^)] **1** and [(Hdap^+^)_2_(tda^2−^)]·2H_2_O **2**. The letters in brackets refer to the symmetry codes shown in the text and figures.

Comp.	D*–*H⋯A	D*–*H	H⋯A	D⋯A	∠DHA	Symmetry Code
**1**	O10–H10⋯O12 ^(a)^	0.84(3)	1.91(3)	2.741(2)	171(2)	−x + 1, −y, −z
	O21–H21⋯O11 ^(f)^	0.91(2)	1.60(3)	2.505(2)	179(2)	x, −y, −z + 1
	N1–H1⋯O11 ^(b)^	0.89(2)	1.88(2)	2.765(2)	171(2)	x, y + 1, z
	N1–H1⋯O12 ^(b)^	0.89(2)	2.38(2)	2.999(2)	127.4(2)	x, y + 1, z
	N6–H6A⋯N7 ^(c)^	0.85(2)	2.11(2)	2.928(2)	163(2)	−x + 1, −y + 1, −z + 1
	N6–H6B⋯O22 ^(d)^	0.88(2)	2.07(2)	2.849(2)	146.9(2)	−x, −y + 1, −z + 1
	C2–H2⋯O12 ^(b)^	0.95	2.52	3.096(2)	118.9	x, y + 1, z
	C8–H8⋯O22 ^(e)^	0.95	2.37	3.233(2)	150.7	x + 1, y, z
	C9–H9A⋯S1 ^(e)^	0.99	3.03	3.748(2)	130.6	x + 1, y, z
	C11–H11A⋯O21 ^(f)^	0.99	2.57	3.319(2)	132.6	−x, −y, −z + 1
	C11–H11B⋯O10 ^(g)^	0.99	2.42	3.365(2)	160.4	−x, −y, −z
	C21–H21A⋯O21 ^(h)^	0.99	2.60	3.375(2)	135.5	−x + 1, −y, −z + 1
	C21–H21B⋯O22 ^(e)^	0.99	2.27	3.252(2)	172.7	x + 1, y, z
**2**	N1–H1⋯O1 ^(b)^	1.08(4)	1.56(4)	2.637(5)	177(4)	x + 1/2, −y + 1/2, z − 1/2
	N2–H2A⋯N7 ^©^	0.93(5)	1.99(5)	2.920(5)	172(4)	x + 1/2, −y + 1/2, z + 1/2
	N6–H6A⋯O11 ^(d)^	0.96(5)	1.92(5)	2.805(5)	152(4)	−x + 1/2, −y + 1/2, −z
	N6–H6B⋯N3 ^(e)^	0.94(5)	2.07(5)	2.991(5)	166(4)	X − 1/2, −y + 1/2, z − 1/2
	N9–H9⋯O11 ^(a)^	0.93(5)	1.79(5)	2.713(5)	175(4)	−x + 1, y, −z + 1/2
	O1–H1A⋯O12 ^(f)^	1.00(10)	1.62(10)	2.607(6)	170(8)	−x, y, −z + 1/2
	O1–H1B⋯O12 ^(g)^	0.79(10)	1.93(10)	2.628(6)	146(10)	x, −y, z + 1/2

**Table 2 ijms-24-17381-t002:** Intermolecular π–π interaction parameters (Å, °) * for [(H9Heade^+^)(Htda^−^)] **1**.

π⋯π	Cg(I)⋯Cg(J)	α
Cg(1)⋯Cg(2) ^(i)^	3.476	0.202
Cg(2)⋯Cg(1) ^(i)^	3.476	0.202
Cg(2)⋯Cg(2) ^(i)^	3.391	0

* Cg(1): ring (N7/C5/C4/N9/C8); Cg(2): ring (N1/C2/N3/C4/C5/C6). Symmetry code: ^(i)^ −x + 3/2, −y + 1/2, −z. Cg(I)⋯Cg(J): Distance between ring centroids; α: Dihedral angle between planes I and J.

**Table 3 ijms-24-17381-t003:** Crystal data and structure refinement for [(H9Heade^+^)(Htda^−^)] **1** and [(Hdap^+^)_2_(tda^2−^)]·2H_2_O **2**.

Compound	1	2
Empirical formula	C_11_H_15_N_5_O_5_S	C_14_H_22_N_12_O_6_S
Formula weight	329.34	486.49
Temperature/K	100(2)	299(2)
Wavelength/Å	0.71073	1.54178
Crystal system	Triclinic	Monoclinic
Space group	*P* $\bar{1}$	*C*2/*c*
Unit cell dimensions		
*a*/Å	4.7676(2)	7.1171(8)
*b*/Å	11.8744(5)	24.422(3)
*c*/Å	13.3350(6)	12.0629(18)
*α*/º	111.563(1)	90
*β*/º	94.144(2)	96.949(8)
*γ*/º	99.129(2)	90
Volume/Å^–3^	686.15(5)	2081.3(5)
*Z*	2	4
Calc. density/Mg/m^3^	1.594	1.553
Absorp. coefc./mm^–1^	0.271	1.949
*F*(000)	344	1016
Crystal size/mm	0.21 × 0.11 × 0.04	0.12 × 0.10 × 0.08
*θ* range/°	2.950–30.504	3.620–67.156
Limiting indices/*h*,*k*,*l*	−6/6, −16/16, −18/18	−7/8, −28/25, −14/13
Refl. collect/unique [R_int_]	31591/4178 [0.0820]	8717/1856 [0.1315]
Completeness *θ*/°, %	25.242, 99.8	67.679, 97.9
Absorp. correct.	Semi-empirical	Semi-empirical
Max./min. transm.	1.000/0.935	1.000/0.773
Data/parameters	4178/214	1856/174
Goodness-of-fit on *F*^2^	1.077	1.019
Final *R* indices	R_1_ = 0.0494, *wR*_2_ = 0.0880	*R*_1_ = 0.0681, *wR*_2_ = 0.1208
*R* indices (all data)	*R*_1_ = 0.0765, *wR*_2_ = 0.1001	*R*_1_ = 0.1360, *wR*_2_ = 0.1483
Largest dif. peak/hole e.Å^–3^	0.537/−0.352	0.409/−0.298
CCDC number	2191905	2191904

## Data Availability

Data are contained within the article and Appendix A.

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
