# Peer review of "Supramolecular Nature of Multicomponent Crystals Formed from 2,2′-Thiodiacetic Acid with 2,6-Diaminopurine or N9-(2-Hydroxyethyl)adenine"

_ijms, 2023, doi:10.3390/ijms242417381_

Round 1

Reviewer 1 Report

Comments and Suggestions for Authors

Summary

The present article deals with the structural investigation of two cocrystals salts made up of a mixture of 2,6-diaminopurine (Hdap) or N9-(2-hydroxyethyl)adenine (9heade) with the conformer 2,2’-thiodiacetic acid (H2tda). The crystal structure of the two multicomponent compound is thoroughly described and shown to both be stabilized by a complex hydrogen bonding network involving both homomolecular and heteromolecular hydrogen bonds. The ionic nature of the constituents of the two cocrystal salts is confirmed by IR spectroscopy measurements and by considering the C-O and C=O distances of the carboxylic/carboxylate groups of H2tda in the crystal structures of both cocrystals salts. Investigation of the thermal stability and behaviour of the two cocrystals salts is carried out. Finally, DFR calculations have been performed by the authors to characterize the intermolecular interactions responsible of the cohesion of the crystal structure of the two cocrystals salts and the results agree pretty well with the solved crystal structures.

My opinion

The submitted manuscript shows a comprehensive structural study of the two cocrystals combining both experimental investigation and simulation in a quite complementary way. All results are thoroughly interpreted and undoubtedly demonstrate the ionic nature of the components of the cocrystals thus showing that they are rather salts. The results are in agreement with the so-called DpKa law announced by the authors. I have noticed however that the quality of English is, in some parts of the manuscript (especially in sections 3.2 to 3.4 of the manuscript and also in supplementary material) relatively poor and I strongly suggest that these parts of the manuscript should be thoroughly revised by the authors for the English. Furthermore, some interpretations of the DSC curves are, to my opinion, questionable (see below). For all these reasons, I suggest that this manuscript should be accepted after some minor revisions are achieved by the authors.  

General questions and comments.

1. As said before, my main remark is about regular English mistakes in the manuscript which occurs more starting from section 3.2 but not only. Confusion between “to” and “of”, “an” and “a”, “to” and “with” (we say for example “in accordance with” and not “in accordance to”), “this” and “these” (we say “these data” and not “this data”), “board” and “broad” are regularly made. Misuse of “the” (especially when the plural form is used) is also regularly done by the authors. Concerning the interpretation of TGA curves a confusion between “evolving” and “involving” is also systematically made (unless I have not understood what the authors meant). The authors are thus asked to carefully check the entire manuscript and also supplementary material (several grammar and vocabulary mistakes are also found in the latter) for English mistakes.

2. I also have several remarks on the comments and presentation of the DSC curves.

(i) First, authors should indicate in the curves (not “shape” as it is said in page 12 for compound 2 or “behaviour” as it is said for TGA curves in Figure 7) the orientation of the endothermic/exothermic events on their DSC curves (with an arrow for example).

(ii) Second, temperature of phase transition are preferably given as the onset temperature rather than the peak temperature since it is considered that the transition starts at the onset temperature rather at the peak temperature.

(iii) Third, I do not agree with the assignment of the two endothermic broad peaks following the melting peak of compound 1 (figure S7) as polymorphic transitions. Indeed, if they were polymorphic transition, an exothermic peak should have been observed just after the melting peak accounting for the crystallization of a new polymorphic form that would subsequently transform to another form and melt. This peak is not observed. According to the TGA curve, it seems more probable that those peaks might be due to the decomposition of compound 2 after/during the melting. The authors should revise this part.

(iv) You say for compound 2 that the peak of dehydration is observed at around 73 °C (which is consistent with the TGA - IR results) which is then followed by the evaporation of the water released at 100 °C. I do not agree with this statement either. The dehydration temperature is 73 °C so water leaves the crystal structure at this temperature and is evaporated at the very same temperature. I think the small endothermic peak at 100 °C originate from another transition (maybe a polymorphic transition). Whatever, the DSC curve is quite complex and it is not advised to try to interpret all thermal events without a strong evidence from additional measurements (for example temperature-resolved X-ray diffraction).

3. There seems to be a difference between the method of preparation of the crystals given in section 2.3 and what is said in sections 3.1.1. and 3.1.2. Indeed, in the latter sections, it is said that both mixtures have been made from a 2:1 ratio of H2tda and Hdap or 9heade while in section 2 the mass taken for each constituent correspond to an equimolar mixure in both cases. The authors should revise that.

4. The authors should also check the figures since sometimes some subscripts are missing either in the figure (Figure 3 for example) or its caption (Figure 4 for example) or sometimes elements mentioned in the text are missing in the Figure (like arrows in Figure 8b that are mentioned at page 13 line 459).

Page-by-page corrections / questions (typos)

-Page 1, line 28 (Abstract): “… an strong …” should be replaced by “… a strong …”.

-Page 2, Figure 1 (Introduction): “dap” should be replaced by “Hdap”. The use of “dap” is also made somewhere else in the manuscript. The authors should correct it.

-Page 3, line 55 (Discussion and results): The authors should be homogeneous in the writing of the vibration bands in the IR spectra. They should systematically insert a space between the vibration symbol and the bracket which precedes the wave number.

-Page 12, line 398 (Discussion and results): “by two weak at 203.9 °C …” should be replaced by “by two weak peaks at 203.9 °C …”.

-Page 13, line 434 (Discussion and results): “… and tetramer of (b) of compound 1” should be replaced by “… and tetramer of (b) compound 1”.

-Page 15, lines 494 and 502: O-H… N and O-H…H hydrogen bonds while the first in not observed in any of the two crystal structures shown in the manuscript and the second just do not exist. The authors should check those mistakes.

Comments on the Quality of English Language

See comment 2 above

Reviewer 2 Report

Comments and Suggestions for Authors

The paper by Belmont-Sánchez et al. describes the synthesis and the nature of two multicomponent crystals, characterize their structure and hydrogen-bond patterns. The study are a complex work, involving different methods applied for the compound characterization (X-ray, calorimetry, spectra and theoretical calculations). All the applied methods allowed to determine the form of the obtained multicomponent crystals (salt/co-crystal problem), which is an important issue, especially in case of the pharmaceutical compounds. These kind of study are need, because the obtained information about particular compounds and their preferences for multicomponent crystal formation can be helpful in future studies of them

I recommend this paper for being published in “International Journal of molecular Science” after some minor revision:

Figure 1: the chemical diagrams of the studied compounds should be in the same scale;

For the studied compounds numbers 1 and 2 (bold) are used, however sometimes they are not bold or they are in brackets. Please use the same code in the article;

Page 5: there is “H2dta” instead of “H2tda”;

Figure 3: in the figure caption there are references for c), d), e) and f), however they are not included in the figure;

In the text some of hydrogen bonds are described as “new”. For example: “Both dimeric units are connected to each other through new hydrogen bonds.” I understand that “new” means that they are different from the ones joining the anions in this case, however maybe it would be better to use “another” or “different” or “additional” hydrogen bond instead of “new”, suggesting that a new type of hydrogen bonds has been discovered;

Figure 5: seems to be a repetition of Figure 4 as the part 5a can be clearly visible in Figure 4a and part 4b does not provide new information. I suggest to remove it as the structure and hydrogen-bond patterns of compound 2 are well described/visible in Figures 4 and 6;

Figures 8-11 could be in better quality;

Conclusions: the obtained compound are considered in the article as salts after the analyses of the obtained results, however in the “Conclusions” they are named “co-crystals”. Why?

Reviewer 3 Report

Comments and Suggestions for Authors

Jeannette Carolina Belmont-Sánchez et al. have submitted a manuscript investigating Supramolecular nature of multicomponent crystals formed 3 from 2,2’-thiodiacetic acid with 2,6-diaminopurine or 4 N9-(2-Hydroxyethyl)adenine.   

While the findings can be interesting, they are not presented in a very good or balanced way. The authors have done tremendous work to support their investigation which I immensely appreciate but in its present state, the overall quality of the work is average. Upon major revision, it may be suitable for publication. Below are some specific comments I hope can aid the authors in improving the manuscript.

1)  I would urge authors to do the NMR (both proton and carbon) and ESI-MS analysis as it will increase the impact of this very nice paper. Please write a paragraph about NMR and ESI-MS analysis in result and discussion section.

2) Instead of ball and stick model of crystal structure of 1 and 2 author should use thermal ellipsoid model.

3) Why the authors chooses under room temperature crystallographic data collection and structure determination.

Overall my assessment is that this paper deserves publication in IJMS after addressing abovementioned comments and thus it will attract audience interested in such type of supramolecular materials.

Round 2

Reviewer 3 Report

Comments and Suggestions for Authors

The manuscript can be accepted in its current form.